# Potential Therapeutic Use of Aptamers against HAT1 in Lung Cancer

**DOI:** 10.3390/cancers15010227

**Published:** 2022-12-30

**Authors:** José Ignacio Klett-Mingo, Celia Pinto-Díez, Julio Cambronero-Plaza, Rebeca Carrión-Marchante, Miriam Barragán-Usero, María Isabel Pérez-Morgado, Eulalia Rodríguez-Martín, María del Val Toledo-Lobo, Víctor M. González, Maria Elena Martín

**Affiliations:** 1Grupo de Aptámeros, Departamento de Bioquímica-Investigación, IRYCIS-Hospital Universitario Ramón y Cajal, Carretera de Colmenar Viejo Km. 9.100, 28034 Madrid, Spain; 2Aptus Biotech SL, Av. Cardenal Herrera Oria 298, 28035 Madrid, Spain; 3Departamento de Inmunología, IRYCIS-Hospital Universitario Ramón y Cajal, Carretera de Colmenar Viejo Km. 9.100, 28034 Madrid, Spain; 4Unidad de Biología Celular, Departamento de Biomedicina y Biotecnología, Universidad de Alcalá, 28871 Alcalá de Henares, Spain

**Keywords:** aptamer, SELEX, HAT1, therapeutic target, inhibition

## Abstract

**Simple Summary:**

Lung cancer is one of the leading causes of death worldwide and the most common of all cancer types. Histone acetyltransferase 1 (HAT1) has attracted increasing interest as a potential therapeutic target due to its involvement in multiple pathologies, including cancer. Aptamers are single-stranded RNA or DNA molecules whose three-dimensional structure allows them to bind to a target molecule with high specificity and affinity, thus making them exceptional candidates for use as diagnostic or therapeutic tools. In this work, aptamers against HAT1 were obtained, subsequently characterized, and optimized, showing high affinity and specificity for HAT1 and the ability to inhibit acetyltransferase activity in vitro. Of those tested, the apHAT610 aptamer reduced cell viability, induced apoptosis and cell cycle arrest, and inhibited colony formation in lung cancer cell lines. All these results indicate that the apHAT610 aptamer is a potential drug for the treatment of lung cancer.

**Abstract:**

Lung cancer is one of the leading causes of death worldwide and the most common of all cancer types. Despite scientific advances in recent years, which have led to the development of new diagnostic and therapeutic approaches, the search for new treatments and early diagnosis of the disease remains an absolute necessity in oncology. Histone acetyltransferase 1 (HAT1) belongs to the HAT family of enzymes and has attracted increasing interest in recent years due to its involvement in multiple pathologies. HAT1 overexpression is related, among other outcomes, to viral infections, inflammatory diseases, and cancer, where it is associated with poor prognosis and low survival. Therefore, many authors propose HAT1 as a potential therapeutic target. Aptamers are single-stranded RNA or DNA molecules whose three-dimensional structures allow them to bind to a target molecule with high specificity and affinity. This makes them exceptional candidates for use as diagnostic or therapeutic tools, among other applications. In this work, aptamers against HAT1 were obtained from oligonucleotide libraries using the systematic evolution of ligands through the exponential enrichment (SELEX) approach. After six rounds of screening, two specific aptamers were obtained and subsequently characterized and optimized. Both aptamers and one derivative based on modified sequences recognized HAT1 with high affinity and specificity and were able to inhibit the acetyltransferase activity of HAT1 in vitro. Furthermore, application of the apHAT610 aptamer resulted in reduced cell viability, induced apoptosis and cell cycle arrest, and inhibited colony formation in lung cancer cell lines. In addition, the apHAT610 aptamer inhibited HAT1 activity in these three cell lines, thus promoting a decrease in histone H4 acetylation and HAT1 protein levels. All these results indicate that the apHAT610 aptamer is a potential drug for the treatment of lung cancer.

## 1. Introduction

Lung cancer is one of the leading causes of death worldwide [1,2] and the most common among all types of cancer [3]. Scientific advances in recent years have allowed the development of new diagnostic and therapeutic approaches that have, in many cases, contributed to improving the prognosis of patients and the practice of personalized medicine, in which the therapeutic decision depends on the characteristics of the tumor in each patient. In recent years, a series of proteins whose levels are altered in different pathologies have been discovered, representing indicators of disease [2] and new and hopeful therapeutic targets. Although it has been made possible to significantly increase the time to tumor recurrence through advancements, lung cancer is still lethal. It is, therefore, essential to continue the search for new targets to counteract this disease.

Histone acetyltransferase 1 (HAT1) is a B-type HAT [4,5] included in the GNAT superfamily. First discovered in *S. cerevisiae* [6], it has since been found in different organisms, including both plants and animals [7,8]. There are currently two ways in which HATs are currently classified. The first is based on their localization and the main substrate they recognize, with type-A HATs involved in recognition of nucleosomal histones and located exclusively in the nucleus, and type-B HATs involved in acetylating free histones and found in the cytoplasm [9,10]. The second classification, based on their structural homology and substrate binding, has been developed in a complementary manner to the previous one to facilitate understanding of this group of enzymes. This involves classification according to three superfamilies of HATs: GNAT (GCN5-Related N-AcetylTransferases), p300/CBP (CREB Binding Protein), and MYST (named after its founding members: MOZ, Ybf2/Sas3 (Yeast), Sas2, and Tip60). Beyond these three superfamilies, some transcription factors, coactivators, and nuclear receptors also exhibit HAT activity [9,11].

Initially, HAT1 was characterized by acetylating free histones and being found in the cytoplasm, hence its classification as a type-B HAT. However, it is now known that HAT1 acetylates other nonhistone substrates [12,13] and is mostly localized in the nucleus [8,14,15] and, to a lesser extent, the cytoplasm [10], and it even presents mitochondrial localization [16]. The main function of HAT1 is to acetylate lysines 5 and 12 of cytoplasm-localized histone H4 for trans transport to the nucleus [8,10,15,17].

The molecular characterization of HAT1 is well-described, but its functions and mechanisms are still poorly understood [5]. In recent years, this enzyme has gained much importance due to its involvement in multiple pathologies, such as inflammatory diseases, viral infections [13,18,19,20,21,22,23,24,25,26,27,28], and cancer. The overexpression of HAT1 at the mRNA and/or protein level in tumor tissue with respect to nontumor tissue is a feature of most studied tumor types, such as lung cancer [29], pancreatic ductal adenocarcinoma [30], prostate cancer [31], diffuse large B-cell lymphoma, peripheral nonspecific T-cell lymphoma and extranodal nasal-type T/NK-cell lymphoma [32], primary esophageal cancer tumors and adjacent tissues [33], actinic myelitis and squamous cell carcinoma of the lip [34], nasopharyngeal cancer [35], osteosarcoma [36], uterine leiomyosarcoma and smooth muscle tumor of uncertain malignant potential [37], cervical cancer [38], liver cancer [39,40], and in both primary tumors and metastases of colorectal cancer [41]. Furthermore, this increase in HAT1 levels is associated with poor prognosis, tumor differentiation, and survival of cancer patients [30,32,33,37].

HAT1 depletion in lung adenocarcinoma and liver cancer [42], pancreatic cancer [30], osteosarcoma [36], and cervical cancer [38] tumor cells causes a decrease in cell proliferation and colony formation and promotes tumor cell apoptosis. In esophageal cancer cells, cell arrest is induced in the G2/M phase of the cell cycle [33]. In murine models of pancreatic cancer [30], subcutaneously injected lung cancer cells [43], prostate cancer xenograft [31], osteosarcoma [36], breast cancer [42], and liver cancer [39], this depletion or inhibition results in a decrease in tumor size.

Although the real relevance of this enzyme in cancer remains to be determined, for which more studies are needed, the available data are sufficient to indicate that HAT1 can be considered as a potential therapeutic target [9,33,35,39,44,45]. In addition, we have analyzed HAT1 levels with immunohistochemistry in different types of cancer, and our results show a statistically significant overexpression of HAT1 in lung, brain, breast, prostate, and bladder tumor tissues compared with nontumor tissues; this overexpression is correlated with poor differentiation and more aggressive, invasive, and metastatic tumors with poorer prognoses (unpublished results).

Two specific HAT1 inhibitors have recently been developed [43,46], and there have been no studies performed on these aside from those of the authors who synthesized and initially characterized them. Therefore, it is necessary to develop more specific inhibitors that contribute to the study of this enzyme and that can serve as a starting point for the development of new therapeutic tools.

Aptamers are single-stranded RNA or DNA oligonucleotides whose three-dimensional conformations facilitate binding to a target molecule with high specificity and affinity [47]. They were discovered by two independent research groups in 1990 [48,49] and have since attracted considerable academic and industrial interest given their immense potential as diagnostic and therapeutic tools [50,51,52,53]. The procedure by which these molecules are selected is known as SELEX (systematic evolution of ligands by exponential enrichment) [49]. This technique consists of performing several rounds of selection in which an initial library of aptamers is presented to the molecule of interest, to which the sequences that recognize it are joined and the others are discarded. As the rounds progress, the conditions become more restrictive such that the oligonucleotides that are most similar and specific to the target are obtained at the end of the process.

The defining characteristics of aptamers make them highly useful in biomedicine because they are molecules demonstrating high stability to changes in temperature and pH as well as in their environment [47,54,55]. In addition, their dissociation constants are in the nanomolar range [56,57], they generate low immunogenicity and toxicity [47,58], have high reproducibility since they do not present batch-to-batch variations, and are low in cost, especially DNA aptamers [59].

In this study, we selected and characterized DNA aptamers that recognize HAT1 with high affinity and specificity and are able to inhibit the acetyltransferase activity of HAT1 in vitro. Of these, the apHAT610 aptamer shows therapeutic activity by reducing cell viability, inducing apoptosis and cell cycle arrest, and inhibiting colony formation in lung cancer cell lines. In addition, this aptamer is also shown to inhibit the acetyltransferase activity of HAT1 in these lung cancer cells.

## 2. Materials and Methods

### 2.1. Materials

Synthetic random ssDNA containing a central region of 40 randomized nucleotides flanked by two conserved 18-nucleotide regions (5′-GCGGATGAAGACTGGTCT-40N-GTTGCTCGTATTTAGGGC-3′), named RND40, and oligonucleotides (Appendix A) were purchased from IBA Life Sciences (Goettingen, Germany). The origin of the remaining material used in this study is indicated within the text.

### 2.2. Expression and Purification of Recombinant HAT1

Human HAT1 Gene ORF cDNA was subcloned in the pQE30 expression vector from pGEM-T Vector (Sinobiological, Eschborn, Germany) and purified by affinity chromatography using an AKTA prime plus system (GE Healthcare, Madrid, Spain) as described [60]. Briefly, Rosetta bacteria expressing recombinant HAT1 were harvested and resuspended in binding buffer containing 20 mM Tris HCl pH 7.8, 0.5 M NaCl, 0.02 M imidazole, and then incubated with lysozyme (1 mg/mL) for 30 min at 4 °C. Cells were then lysed by sonication and centrifuged at 10,000× *g* for 30 min at 4 °C, and the lysate was loaded onto a HisTrap FF column (GE Healthcare, Madrid, Spain). Recombinant HIS-HAT1 was eluted with 0.5 M imidazole (Appendix A). Purified HIS-HAT1 was aliquoted and frozen at −80 °C. Protein concentration was determined using a BCA kit (PierceTM ThermoFisher Scientific, Waltham, MA, USA).

### 2.3. In Vitro Selection

Iterative rounds of selection and amplification of ssDNA aptamers were performed as previously described by Morris et al. [61]. In brief, 1 nmol of the initial library RND40, diluted in selection buffer (PBS (Gibco, New York, NY, USA), 1 mM MgCl_2_), heated at 95 °C for 10 min and cooled on ice for 10 min, and then incubated with 2 µg (50 pmoL) of recombinant His-HAT1 at 37 °C for 1 h or 30 min from the fourth round of selection. Next, 50 µL of Ni-NTA Superflow (Qiagen, Madrid, Spain) was added, incubated for 1 h at 4 °C, and then washed three times with 1 mL of selection buffer. The ssDNA–protein complexes were resuspended in 100 µL of hot PCR Mix (90 °C) containing 0.8 µM/primers F3 and R3 (Appendix A), 200 µM dNTPs, 3 mM MgCl_2_, and 1 U of Taq polymerase (Biotools, Madrid, Spain). The PCR was performed for 10 cycles. The oligonucleotides were purified using NucleoSpin Gel and a PCR Clean-Up kit (Macherey-Nagel, Düren, Germany) according to the manufacturer’s instructions.

### 2.4. Aptamer Cloning, Sequencing, and Secondary Structure Prediction

The aptamers obtained in round 6 of selection were cloned using pGEM-T Easy vector systems (Promega, Madison, WI, USA) according to the manufacturer’s instructions, and individual clones were sequenced using the primers T7 and SP6 (Appendix A). Predicted secondary structures were obtained using the mFold software (http://mfold.rna.albany.edu/?q=mfold/DNA-Folding-Form) accessed on 4 April 2019 with the following parameters: 37 °C, 150 mM [Na^+^], and 1 mM [Mg^+2^]. QGRS Mapper, a web-based server, was used to predict G-quadruplexes in the selected aptamers.

### 2.5. Next-Generation Sequencing

Aptamer populations from rounds 3 and 6 were sent to the Genomic Unit of the National Center for Cardiovascular Research (CNIC) to perform next-generation sequencing (NGS). The files that resulted from the sequencing were analyzed to check the evolution of the population in the aptamer selection by using Python software to perform sequence alignment and, thus, to be able to see similarities within each population. The sequences were first ordered and adjusted to the same orientation (forward), and screening was then performed such that duplications and concatemers of primers were avoided.

### 2.6. Enzyme-Linked Oligonucleotide Assay (ELONA)

Recombinant His-HAT1 protein or cell lysate (100 ng) diluted in coating buffer (Seracare, Biogen Cientifica, Madrid, Spain) was plated in a microtiter plate of 96 wells (NUNC, ThermoFisher Scientific, Massachusetts, USA) and incubated overnight at 4 °C. The plate was then blocked using BSA 0.2% in selection buffer and incubated for 1 h at room temperature, followed by washing the plate three times in PBS-Tween 0.1%. The aptamers conjugated with biotin and digoxigenin were diluted in selection buffer to prepare various solutions of different concentrations, as indicated in the figure legends, and structured by first heating for 10 min at 95 °C and then incubating for 10 min on ice. Next, 100 µL of the diluted aptamers were added, and the plate was incubated at 37 °C for 1 h. After three washes with PBS-Tween 0.1%, the plate was incubated with 100 μL of a 1/1000 dilution of streptavidin conjugated with horse radish peroxidase (HRP) (Sigma Aldrich, Madrid, Spain) or HRP conjugated anti-digoxigenin antibody (Roche, Madrid, Spain) for 1 h of incubation at room temperature. Finally, the plate was washed as mentioned above and developed using ABTS solution (Roche, Madrid, Spain). The absorbance was measured at 405 nm in an Infinite F200 spectrophotometer (TECAN, Männedorf, Switzerland).

### 2.7. Evolution Analysis of the Aptamer Population by qPCR

qPCR of the initial library (RND40) and all rounds of selection was performed using iQ5 equipment (BioRad, Barcelona, Spain). Quantitative analysis was performed using a Quantimix Easy Kit (Biotools, Madrid, Spain), F3 and R3 oligonucleotides under the conditions of 0.03 µM of each primer, and 2 µL PCR product (approximately 1 ng) in a final volume of 20 µL at 57 °C for 25 cycles. The melting curve was 55 °C for 81 cycles.

### 2.8. In Vitro HAT1 Activity Assay

The in vitro assay for acetyltransferase activity of His-HAT1 was performed with or without aptamers. His-HAT1 (100 ng or 2.5 pmol) was incubated either with the structured aptamers in the selection buffer or only with the selection buffer for 30 min at 37 °C. Next, 10 µL of Ni-NTA Superflow was added and incubated for 1 h at 4 °C. After three washes with the selection buffer, 27 µL of reaction buffer (100 mM Tris-HCl pH 7.6 and 250 ng of histone H4 (Abcam, Cambridge, UK)) was added and incubated for 1 min at 37 °C. Reaction began with the addition of 3 µL AcCoA 10 mM (Roche, Madrid, Spain) in 30 µL of final volume. Finally, 15 µL of denaturing buffer (Tris-HCl 180 mM pH 6.8, SDS 9%, β-mercaptoethanol 6%, glycerol 15%, and bromophenol blue 0.025%) was added to stop the enzymatic reaction after 2 h of incubation.

### 2.9. Aptamer Stability Assays

The aptamers (3.6 µg) were incubated with 60 µL of human plasma at 37 °C up to 72 h. Aliquots were extracted from the mixture at 0, 6, 24, 48, and 72 h, then heated for 10 min at 65 °C to inhibit the activity of the nucleases present in the plasma, and then cooled on ice for 10 min. They were kept at −20 °C until all the aliquots were obtained, which were then loaded on a 3% low-binding agarose gel.

### 2.10. Cell Culture and Protein and RNA Extraction

The three cell lines used in this study, generously donated by Dra. Eloisa Jantus, were authenticated using the GenePrint^®^ 10 System (A549 and SW900 in May 2019 and H1650 in June 2020).

A549 cells were maintained in a growth medium (Dulbecco’s modified Eagle’s medium (DMEM) (Biowest SAS, Nuaillé, France) containing 10% fetal calf serum (Gibco, New York, NY, USA), 1% L-glutamine and 100 U/mL penicillin, 100 µg/mL streptomycin, and 25 µg/mL amphotericin (Sigma, St. Louis, MO, USA)) in a humidified 5% CO_2_/95% air incubator at 37 °C.

SW900 and H1650 cells were maintained in a growth medium (RPMI (PAA, Pasching, Austria) containing 10% fetal calf serum, 1% pyruvate, 1% glutamine and 100 U/mL penicillin, 100 µg/mL streptomycin, and 25 µg/mL amphotericin) in a humidified 5% CO_2_/95% air incubator at 37 °C.

To obtain cell lysates, cells were detached using Trypsin/EDTA, resuspended in culture medium, and centrifuged at 400× *g* for 5 min. The cells were lysed with RIPA buffer (50 mM Tris-HCl, pH 7.4; 1 mM EDTA; 150 mM NaCl; 0.5% sodium deoxycholate, 0.1% SDS; 1% NP 40) in a 1:4 volume ratio and frozen at −80 °C for 24 h. Cells were then thawed on ice and sonicated by performing three 15-s pulses at 10-s intervals without sonication. Finally, they were centrifuged at 12,000× *g* for 10 min, and the supernatant (cell lysate) was collected to determine the protein concentration of the sample using the BCA kit. It was then aliquoted and stored at −80 °C until use.

Total RNA from cells was lysed and homogenized in NucleoZOL (Macherey-Nagel, Düren, Germany) by following the manufacturer’s protocol. The RNA pellet was resuspended in 50 μL of RNase-free water and then quantified and stored at −80 °C. From 1 µg of the total extracted RNA, cDNAs were obtained in a final volume of 20 µL using the SensiFAST^TM^ cDNA synthesis kit (Bioline, Barcelona, Spain) by following the manufacturer’s protocol. Subsequently, qPCR was carried out in a Step One Plus^TM^ thermocycler with the AceQ qPCR SYBR^®^ Green Master Mix—Vazyme kit (Quimigen, Madrid, Spain). As a control, the levels of β-actin mRNA were measured. The oligonucleotides used are shown in Appendix A. The reaction was performed in a final volume of 10 µL containing Mix FastGene^®^ IC Green, the oligonucleotide pairs at a final concentration of 0.2 µM, and 2 µL of cDNA at 60 °C for 45 cycles of the PCR protocol with melting.

To quantify mRNA levels, the value of 2^−ΔΔCt^ was calculated using the following formula, where β-actin is the reference gene:ΔΔCt = ΔCt_transfected cells with apHAT610 or control aptamer (20AT)_ − ΔCt_control cells_
ΔCt = Ct_HAT1 gene_ − Ct_reference gene_

### 2.11. Western Blot and Apta-Western Assays

Cell lysates (10–20 μg) were resolved in 15% sodium dodecyl sulfate polyacrylamide gel electrophoresis (SDS-PAGE) and transferred onto PVDF membranes, which were then incubated with either the appropriate antibodies overnight at 4 °C or biotin-structured aptamer for 1 h at 37 °C. After washing, blots were incubated with the corresponding HRP-conjugated secondary antibody (GE Healthcare, Madrid, Spain) or HRP-streptavidin (Sigma Aldrich, Madrid, Spain) for 1 h at room temperature. Finally, the membranes were developed with either the enhanced chemiluminescence (ECL) kit (GE Healthcare, Madrid, Spain) or Clarity Western ECL Substrate (BioRad, Barcelona, Spain). PageRuler Plus Prestained Protein Ladder (Thermo Scientific, Massachusetts, USA) was used in all of the experiments. β-Actin (Sigma, St. Louis, MO, USA) and α-histone H4 (Abcam, Cambridge, UK) antibodies were used as loading controls. Appendix A shows the antibodies used in this study.

### 2.12. Cell Viability (MTT) and Cell Death (LDH) Assays

Cells plated in 96-well plates (10^4^ cells/well) were transfected with the aptamers, after 16–24 h, at the concentrations indicated in the figure legends using Lipofectamine^TM^ 2000 (Invitrogen, Boston, MA, USA) and following the instructions for siRNA transfection. The entry of the aptamers into the cells was verified with confocal microscopy (Appendix A). After 48, 96, or 120 h (A549, H1650, and SW900, respectively), the plates were incubated for 1.5–3 h at 37 °C with MTT (Sigma, St. Louis, MO, USA) diluted 1 mg/mL in culture medium. Next, the cells were lysed with 10% SDS and 10 mM HCl overnight at 37 °C. The absorbance was read at 540 nm in an Infinite F200 spectrophotometer (TECAN).

To perform LDH assays 48, 96, or 120 h after aptamer transfection (A549, H1650, and SW900, respectively), the supernatants were collected to be mixed with the reagent of the Cytotoxicity Detection kit (LDH) (Roche, Madrid, Spain) by following the manufacturer’s protocol. The reaction was incubated for 30 min at room temperature and was stopped by adding 1 M HCl. The solubilized formazan was then measured in an Infinite F200 spectrophotometer at 490 nm. To calculate the percentage of cytotoxicity, two controls were included: the supernatants of the untreated cells (LDH_low_) and of the lysate cells (LDH_high_) with 0.2% Triton X-100. Finally, the results are expressed as cytotoxicity (%) = ((LDH_treated_ − LDH_low_)/(LDH_high_ − LDH_low_) × 100 for each experimental point respective to the control.

### 2.13. Flow Cytometric Analysis of Cell Cycle

At 24 and 72 h post-transfection, the cells were collected and fixed in 70% cold ethanol for 30 min at −20 °C. Subsequently, the cells were washed twice with PBS and incubated with propidium iodide at 50 µg/mL (Sigma, Missouri, USA) and RNAse A at 100 µg/mL (Sigma, St. Louis, MO, USA) for 1 h at 37 °C in darkness. The cell cycle was analyzed with the cytometer (FACSCanto IITM, BD Biosciences, Franklin Lakes, NJ, USA), and doublets of cells with selective gating were excluded. The data were analyzed using the software Flowing 2.5.1 (Turku Bioscience, Turku, Finland).

### 2.14. Colony-Forming Assays

Cells transfected with the aptamers were collected after 16–24 h, and live cells were counted according to the trypan blue exclusion assay (Sigma, St. Louis, MO, USA) using the counter TC20 (BioRad, Barcelona, Spain). Cells were seeded at 1 × 10^3^ cells/well in 6-well plates. After 6–12 days, the colonies were fixed with 1 mL/well methanol (Sigma Aldrich, Madrid, Spain) for 10 min at room temperature, stained for 30 min with Giemsa 0.02% (Sigma, St. Louis, MO, USA), and images were taken in the image analyzer ChemiDoc MP Imaging Systems (BioRad, Barcelona, Spain) to be counted using ImageJ software version 1.8.0_172 (National Institute of Health, Bethesda, MD, USA).

### 2.15. Statistical Analysis

Data are presented as an average value ± standard error of the mean (SEM). To determine if the differences between the averages of the different experimental groups were significant, the statistical analysis was performed through an ANOVA test followed by Tukey’s test or one-sample t-test against a control value with GraphPad Prism 8.0 (San Diego, CA, USA). Significance was assumed at *p* < 0.05.

## 3. Results

### 3.1. Selection and Characterization of High-Affinity Aptamers against HAT1

Six successive rounds of selection were carried out using a Ni-NTA resin to which the protein was bound because it was fused to a 6xHis tail to facilitate purification. To discover the evolution of the selection and to confirm the enrichment of the obtained populations, ELONA assays were performed. As shown in Figure 1, the results show a statistically significant increase in the signal in the populations obtained in rounds three and six with respect to the initial population (RND40) (7- and 15-fold, respectively) and between rounds three and six (2-fold).

As an alternative method to ELONA that would allow us to see the evolution of the population, a real-time PCR assay was performed to analyze the enrichment of the population after six selection rounds. There was a drop in fluorescence in the initial rounds due to the great variability in sequences, which decreases as the selection progressed and there was increasing enrichment of the target; this caused fluorescence to increase, which indicates that selection was successful, as described by Luo and collaborators. The same occured with the melting curve of these assays, where there was an evolution toward higher Tm and higher fluorescence peaks for successive rounds with less sequence variability [62]. As can be seen in Figure 2, the assay evolution indicates a decrease in sequence variability with each progressive round of selection, thus resulting in an enrichment of aptamers against the HAT1 protein.

### 3.2. Obtaining Unique Sequences That Specifically Recognize HAT1

This process was performed in two different ways: (i) round six cloning and subsequent Sanger sequencing, and (ii) next-generation sequencing (NGS) of clones from rounds three and six to study the evolution of the population during selection.

Seven sequences were obtained from cloning and Sanger sequencing. These sequences and their complementary strands were labeled with digoxigenin, and their affinities for HAT1 were measured with ELONA. As shown in Figure 2, the aptamers 63F, 610F, 63R, and 610R had higher absorbance values than the other aptamers and the negative control.

With the second approach, the next-generation sequencing of clones from rounds three and six was performed, and the sequence similarity within each population was analyzed. After screening, there were 396 of the initial 1658 sequences for round 3, and 877 of the 5775 initial sequences for round six were useful for analysis. Figure 3A corresponds to two heat maps depicting sequences from each population grouped by similarity, with lighter colors indicating sequences with higher similarity. In both rounds three and six, there were four distinct groups or clusters which had a high intracluster but low intercluster similarity, and these were maintained throughout the selection. Evolution in favored selection toward the apHAT63 aptamer can also be observed, as its enrichment improved from round three to six to the detriment of apHAT610, which could indicate the higher affinity of apHAT63 for HAT1.

If each cluster is analyzed individually, as shown in the example of the apHAT63 cluster from round three in Figure 3B,C, there was one sequence that was most represented within each cluster, and it coincided with the one that had the highest similarity to the rest of the sequences in its cluster (the lighter color). The intracluster similarity is very high because they only differed from each other by a few nucleotides in the whole sequence. Furthermore, the sequence most represented within the apHAT610 cluster matches the 610-aptamer isolated from Sanger sequencing, and that of the apHAT63 cluster matches aptamer 63, with the exception that the Sanger sequencing had an extra nucleotide at the end of the variable region that was probably added during cloning or sequencing.

In view of the results obtained in NGS, it can be stated that the aptamers, obtained by Sanger sequencing with the highest affinity for ELONA, were the two most represented in NGS in both their direct and inverse forms. Therefore, the apHAT63F, apHAT63R, apHAT610F, and apHAT610R aptamers were chosen for further study (sequences are shown in Appendix A). Since the NGS analysis was performed after cloning and Sanger sequencing, the sequences of the apHAT63F and R aptamers used in this study were those obtained by Sanger sequencing with one extra nucleotide.

### 3.3. Structural Characterization and Optimization of the Selected Aptamers

The structure of the four aptamers was analyzed in silico using the mFold software, which allows prediction of the most stable secondary structures based on their free energy (ΔG). The nucleotide compositions of the aptamers, which is relevant for their stability due to the possible formation of G-quadruplex structures, were also analyzed. Thus, the GC richness (%GC) was calculated, and QGRS Mapper software was used to predict whether the sequences could adopt these structures. The most likely secondary structures of the aptamers apHAT63F, apHAT63R, apHAT610F, and apHAT610R are those that present a lower free energy (Appendix A). Of the four, only two had possible G-quadruplex structures according to the prediction: apHAT63F and apHAT610R (renamed apHAT63 and apHAT610, respectively); thus, aptamers apHAT63R and apHAT610F were excluded from further study. Taking into account these predictions, the possibility arose of truncating the two selected aptamers to eliminate regions that, a priori, would not pose an impediment for their binding to the target. Thus, the last 14 nucleotides of the 3′ end of apHAT63 and the first 14 nucleotides of the 5′ end of apHAT610 were removed (Appendix A). The truncated aptamers, named apHAT63T and apHAT610T, each maintained a similar structure and Gibbs free energy value to those of their respective parental aptamers (Appendix A). Furthermore, the analysis with the QGRS Mapper program indicates that the possible G-quadruplex structures present in both parental aptamers are conserved in their truncated aptamers. All of these results indicate that the new molecules could maintain their activity.

### 3.4. The Aptamers Selected against HAT1 Recognize Their Target with High Affinity and Specificity

To test the affinity and specificity of the aptamers selected for their target, an ELONA assay of the four aptamers against either the HAT1 protein or a cell lysate of MDA-MB-231 breast cancer cells (control) was performed. As shown in Figure 4A, apHAT63, apHAT63T, and apHAT610 recognized their target with high affinity and specificity, whereas they did not recognize the proteins in the cell lysate, and the differences are statistically significant. However, apHAT610T showed lower target recognition than its parental apHAT610 aptamer and was therefore excluded from further study.

To determinate the affinity constant of the selected aptamers apHAT63, apHAT63T, and apHAT610, ELONA assays were performed in which the HAT1 protein was incubated with increasing concentrations of biotin-labeled aptamers. The data obtained were analyzed by nonlinear regression, which showed that they fit a hyperbola curve whose equation is y = (x Bmax)/(x + K_D_), where Bmax is the maximum binding and K_D_ (dissociation constant) is the concentration of aptamers required to reach half of the maximum binding. Thus, the aptamers studied were able to detect the HAT1 protein in a concentration-dependent manner with a K_D_ of 28.11 nM for apHAT610, 38.01 for apHAT63, and 43.71 for apHAT63T (Figure 4B).

Next, apta-Western blots were performed to analyze the affinity of the aptamers by denatured HAT1. apHAT63 and apHAT610 recognition occurred in a concentration-dependent manner, with apHAT610 being the aptamer that best recognized the denatured target at its highest concentration (Appendix A).

### 3.5. Aptamers against HAT1 Inhibit Acetyltransferase Activity In Vitro

The capacity of the three aptamers to inhibit the acetyltransferase activity of HAT1 in vitro was studied as described in the Section 2.

As shown in Figure 4C, the purified HAT1 protein was able to acetylate histone H4, indicating that it had enzymatic activity. Furthermore, the aptamers apHAT63T and apHAT610 inhibited acetylation in a concentration-dependent manner, with an IC50 of 72.24 and 46.61 nM, respectively, which was not observed for the unspecific aptamer, 20AT, used as negative control. Regarding apHAT63, although assays in the presence of this aptamer had high variation, taken together, there was no significant effect on the HAT1 activity.

### 3.6. The Aptamers against HAT1 Are Stable in Human Plasma

The resistance of the aptamers to be degraded by plasma DNases was tested by incubating the three molecules and the control aptamer in the presence of human plasma for 72 h and then obtaining aliquots at several different times. The half-life of the three aptamers was in the range of 26–41 h. The half-life of apHAT63 as compared with its truncated aptamer is 41 ± 7.573 h versus 26 ± 6.539 h, indicating the lower stability of the truncated aptamer (Appendix A).

### 3.7. Aptamers against HAT1 Inhibit Cell Viability in Lung Tumor Cells

The effect of the three aptamers against HAT1 on cell viability was studied in three lung cancer cell lines: A549, H1650 (adenocarcinoma), and SW900 (squamous carcinoma). Cells were transfected with increasing concentrations of either the three aptamers or the nonspecific aptamer 20AT, and MTT activity was measured at twofold doubling time for each line. Thus, the doubling time was 22–28 h for A549 cells, 52 h for SW900 [63], and 40–42 h for H1650 [64]. Therefore, MTT activity was measured at 48 h for A549, at 120 h for SW900, and at 96 h for H1650 (Figure 5). Simultaneously, lactate dehydrogenase (LDH) enzyme activity was measured in the supernatants of the transfected cells at the same times to study possible cell death by necrosis.

As shown in Figure 5A, treatment with all three aptamers decreased MTT activity in all three lines, which allowed the calculation of the IC50 in most cases. The control aptamer produced a slight inhibitory effect, especially in H1650 cells, although less than that produced by the three aptamers. As can be seen in Figure 5B, the reduction in cell viability was not caused by necrosis, as cytotoxicity was null or practically null at the studied concentrations.

### 3.8. apHAT610 Triggers Cell Cycle Arrest in Lung Tumor Cells

The following experiments were conducted using only the apHAT610 aptamer because it decreased MTT activity the most in all three cell lines. From this point, working concentrations were also set that were approximately two times the IC50 for each line, i.e., 20 nM for A549, 45 nM for SW900, and 55 nM for H1650.

At these concentrations, a decrease in the total number of apHAT610-treated cells was observed in all three cell lines, with this decrease being greatest in H1650 cells at 72 h (Appendix A).

The effect of apHAT610 on cell survival was also measured with a clonogenic cell activity assay. Cells were pretransfected with apHAT610 and the control aptamer 20AT, and at 24 h, 1000 cells/well were harvested and reseeded for a colony formation assay. Colony quantification was performed at 6 days for A549 cells, 8 days for SW900 cells, and 12 days for H1650 cells, counted from seeding.

Figure 6A shows the inhibitory action on the clonogenic capacity of apHAT610 cells with respect to the control and 20AT. This inhibition in colony formation was statistically significant for cell lines SW900 and H1650 with respect to the control and for cell lines A549 and H1650 with respect to 20AT.

Next, the effect of apHAT610 on the cell cycle was studied in all three lines. As shown in Figure 6C, apHAT610 aptamer caused a significant reduction of cells in the G1 phase in the SW900 and H1650 cell lines, which was accompanied by an increase of cells in the S and/or G2/M phases, whereas the control aptamer 20AT produced no significant effect in the three cell lines. These results suggest that apHAT610 caused cell cycle arrest in one of the phases (S or G2/M) in these cell lines (Figure 6C).

### 3.9. apHAT610 Induces Apoptosis in Lung Tumor Cells

The activity of caspase 3, a protease activated in apoptosis, was evaluated. A Western blot study was performed on the enzyme poly(ADP-ribose) (PARP), an endogenous substrate of caspase 3, processed in late apoptotic stages. Caspase 3 activity produces characteristic patterns for PARP proteolysis, with specific 89 kDa fragments.

Figure 7 shows the immunodetection of fragments produced by PARP proteolysis in A549, SW900, and H1650 cells 24 h post-transfection with apHAT610 and 20AT aptamers. The results show the apoptotic effect of apHAT610 on the three cell lines, with differences being statistically significant in H1650 cells.

### 3.10. apHAT610 Inhibits the Acetyltransferase Activity of HAT1 in Lung Tumor Cells

Finally, we studied the capacity of apHAT610 to inhibit the acetyltransferase activity in lung tumor cells. For this purpose, cells were seeded and, 24 h later, transfected with apHAT610 or the control aptamer 20AT at the concentrations established above. Twenty-four hours post-transfection, the cells were lysed, and acetylated histone H4 (Figure 8A) and HAT1 protein (Figure 8B) levels were studied by Western blotting.

The results indicate that the apHAT610 aptamer produced a statistically significant decrease in histone H4 acetylation and HAT1 levels in all three cell lines relative to the control, being particularly significant in the SW900 cell line (Figure 8). On the other hand, the inhibition exerted by the 20AT control (around 30% in the three cell lines) could be a nonspecific effect because 20AT did not inhibit HAT1 activity in the in vitro inhibition assays. In addition, the results show that the effect of apHAT610 on HAT1 protein levels was not due to a decrease in the mRNA levels (Appendix A). In SW900 cells, mRNA levels showed a statistically significant increase in apHAT610, which could indicate a compensatory effect in the cell in response to the decrease in protein levels.

## 4. Discussion

Cancer is one of the leading causes of death in the world. Despite advances in the development of treatments and diagnostic tools, it is a disease for which therapeutic options remain limited, and the search for new strategies and biomarkers that facilitate early diagnosis and have prognostic value to aid in decision-making is a priority in oncology.

One of the therapeutic targets that has gained prominence in recent years is HAT1, whose overexpression is known to be related to the development of various pathologies, such as infections caused by viruses (HBV, HIV, and COVID-19) [18,26,27,44] or cancer. The fact that high levels of this protein are observed in cancer may be due to its involvement in different cellular functions during the replication phase when the cell has high levels of HAT1 [10,65]. For example, this enzyme facilitates the availability of acetyl groups in the nucleus, which are transformed into AcCoA even under conditions of nutritional stress [66,67,68,69], thus favoring cell proliferation in aberrant processes. Likewise, its participation in chromatin maturation [70,71] and DNA repair [15,72,73,74,75], its interaction with ORC [65,76], and its decrease in aged versus young tissues [77] show the relevance of this enzyme in cell biology and proliferative processes. Therefore, taking into account that alterations in cell cycle control are one of the characteristics of all types of cancer [78], it is logical to think that increased levels, changes in subcellular distribution, and/or changes in the control of HAT1 activity could be involved in the etiopathogenesis of these pathologies.

In our study, the NGS analysis results indicate the evolution of the population toward four very well-differentiated aptamer groups, at least starting from the third round. This indicates that selection occurs efficiently, as the number of sequences decreases compared with in the initial population of more than 1.5 million sequences (sequence similarity analysis not shown), and that only specific aptamers against the target are selected until round six.

Within each cluster, very high similarities were observed between the comprising aptamers, and in all of them, there was a sequence with the highest percentage of representation that coincided with the one that had the highest similarity with the rest of the sequences in its group [79]. The sequence of the apHAT610 aptamer coincided with that sequence obtained by Sanger sequencing. However, this was not the case with apHAT63 because the aptamer obtained by Sanger sequencing had one nucleotide more than the one most represented in NGS and is not found in either round of NGS. This may be because a mutation occurred in the sequence obtained by Sanger sequencing during the cloning or sequencing process. The aptamer we worked with corresponded to the one obtained by Sanger sequencing because this process was performed before NGS analysis, but this should not preclude further experimental investigation of this sequence because it recognizes its target in an optimal way.

One of the characteristics required when trying to find new antitumor agents is their ability to inhibit certain processes related to tumor development, such as cell proliferation and clonogenic activity. In this sense, this work shows that the three aptamers can reduce cell proliferation in the three studied human lung cancer cell lines (A549, SW900, and H1650) without producing necrosis. However, an IC50 was obtained for all three lines only in the case of the apHAT610 aptamer, and the established values are in the low nanomolar range. In addition, the results of the HAT1 inhibition assays show that apHAT610 inhibited HAT1 activity in all three cell lines and is, therefore, the most promising candidate of all the aptamers. Another important characteristic of the aptamers regarding their potential as anticancer agents is resistance to nuclease degradation. In this respect, the three aptamers showed high resistance to degradation by blood plasma nucleases, showing a half-life greater than 24 h in all cases, which may be due to the possible G-quadruplex structures possessed by all of them [80].

The observed reduction in cell viability may be caused by: (i) necrosis generated by the toxicity of the treatment, which we ruled out with the results obtained in the LDH release assays; (ii) the initiation of apoptosis, which causes the cell to initiate programmed death as it is unable to perform its vital functions; (iii) reduced proliferation due to cell cycle arrest. In the first two processes, treatment results in an irreversible reduction in viability, but in the case of cell cycle arrest, cells can become viable again if treatment is discontinued.

Our results show that there is a reduction in proliferation that may be due to cell cycle arrest in the S and G2/M phases (Figure 6) and/or the induction of apoptosis mediated by caspase 3 (Figure 7). In esophageal squamous cancer cells, HAT1 inhibition has been shown to cause cell cycle arrest in G2/M, and the use of HAT1 as a therapeutic target in this tumor type is proposed [33]. In immortalized mammary epithelial cells, HAT1 depletion induces cycle arrest in G1/S [42]. The inhibition of EGFR leads to cycle arrest in G1 or G2/M phases [81,82]. The combination of one of its inhibitors, erlotinib, with enzastaurin, a PKC-β2 (protein kinase C β2) inhibitor, causes accumulation in A549 and H1650 cells in S and G2/M phases, which is more pronounced in H1650 cells [82]. One of the effects of this synergy is the decrease in the activity of CK2 (casein kinase 2), a kinase that phosphorylates HAT1 [83] and has various functions in the regulation of cell proliferation [84,85]. Our HAT1 inhibitor aptamer presented similar effects to this drug combination, and it would be interesting to analyze whether it affects the phosphorylation of HAT1 by CK2 through blocking the binding site of this kinase. It could be that apHAT610 caused effective arrest at these points of the cell cycle, but it is necessary to perform studies after first synchronizing the cell cycle to confirm these results.

When cells were treated with the apHAT610 aptamer at the stated concentration (2 × IC50), a statistically significant reduction in HAT1 activity, measured by histone H4 acetylation and HAT1 protein levels, was observed in lung cancer cell lines 24 h post-transfection. However, no significant differences in HAT1 mRNA levels in apHAT610-treated cells were observed. In fact, a significant increase in HAT1 mRNA levels wes observed in SW900 cells treated with apHAT610, which could be due to a compensatory effect of the cell.

It can be concluded that the aptamer apHAT610 is an effective inhibitor of histone acetyltransferase 1, both in vitro and in lung cancer cell models. The comparison with the two previously described HAT1 inhibitors shows that apHAT610 aptamer has affinity for its target in the low nanomolar range similar to that of the bisubstrate inhibitor (28.11 nM for apHAT610 and 1.1 nM for H4K12CoA) [46] and inhibits the growth of the three lung cancer cell lines in the nanomolar range in contrast to the micromolar range for the riboflavin analog JG-2016 [43]. These results, together with the fact that aptamers, due to their nucleotide nature, usually show low or no immunogenicity or toxicity, make the apHAT610 aptamer targeting HAT1 a better alternative for further development as an antitumor agent in lung cancer and other tumors.

Despite all these data, much remains to be known about the role of the HAT1 enzyme and how it is affected by the apHAT610 aptamer. Therefore, it is necessary to continue in-depth research to discover new functions and new limitations of this therapeutic target to effectively control its regulation in the treatment of different pathologies.

## 5. Conclusions

Four DNA aptamers (apHAT63, apHAT610, and their truncated sequences) were selected, optimized, and characterized, and three of these recognized HAT1 with high affinity and specificity. In addition, apHAT63T and apHAT610 demonstrated inhibitory effects on HAT1 activity in vitro. The apHAT610 aptamer inhibited cell viability, induced apoptosis and cell cycle arrest, and inhibited colony formation in lung cancer cell lines. Finally, we also showed that apHAT610 inhibited HAT1 activity in vivo. These findings strongly support use of the apHAT610 aptamer as a potentially effective drug in lung cancer therapy.

## Figures and Tables

**Figure 1 cancers-15-00227-f001:**
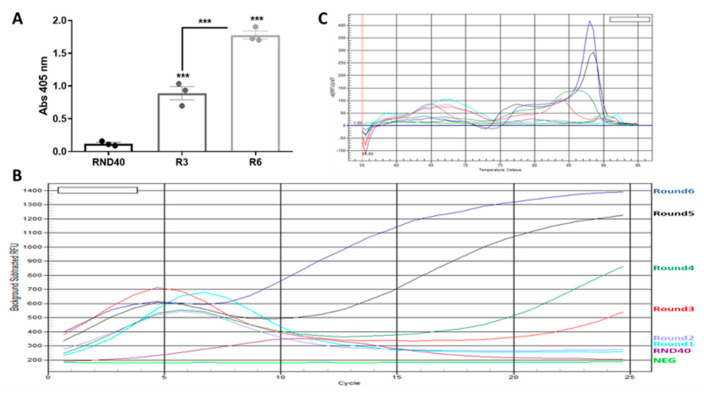
Study of selection evolution. (**A**) ELONA from rounds three and six. Plates with 100 ng/well (2.5 pmol/well) of HAT1 protein bound to them were incubated with 200 ng/well (80 nM) of either the starting population (RND40), round three (R3), or round six (R6) labeled with digoxigenin. The bars show the mean ± SEM. Data shown correspond to three independent experiments (*** *p* < 0.001 of R3 and R6 relative to the initial population, and of R6 relative to R3). (**B**) Real-time PCR and (**C**) melting curve of the initial population (RND40) and the six rounds of selection. The color legend matches the previous image.

**Figure 2 cancers-15-00227-f002:**
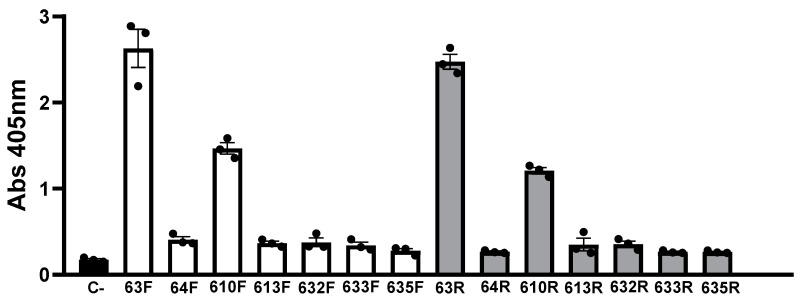
Study of the affinity of the F and R chains of aptamers. Plates with 100 ng/well (2.5 pmoL/well) of HAT1 protein bound to them were incubated with 100 ng/well (40 nM) of each aptamer (63F, 64F, 610F, 613F, 632F, 633F, 635F, 63R, 64R, 610R, 613R, 632R, 633R, and 635R) labeled with digoxigenin. The negative control (C-) carried no aptamer. The bars show the mean ± SEM. The data shown are from three independent experiments.

**Figure 3 cancers-15-00227-f003:**
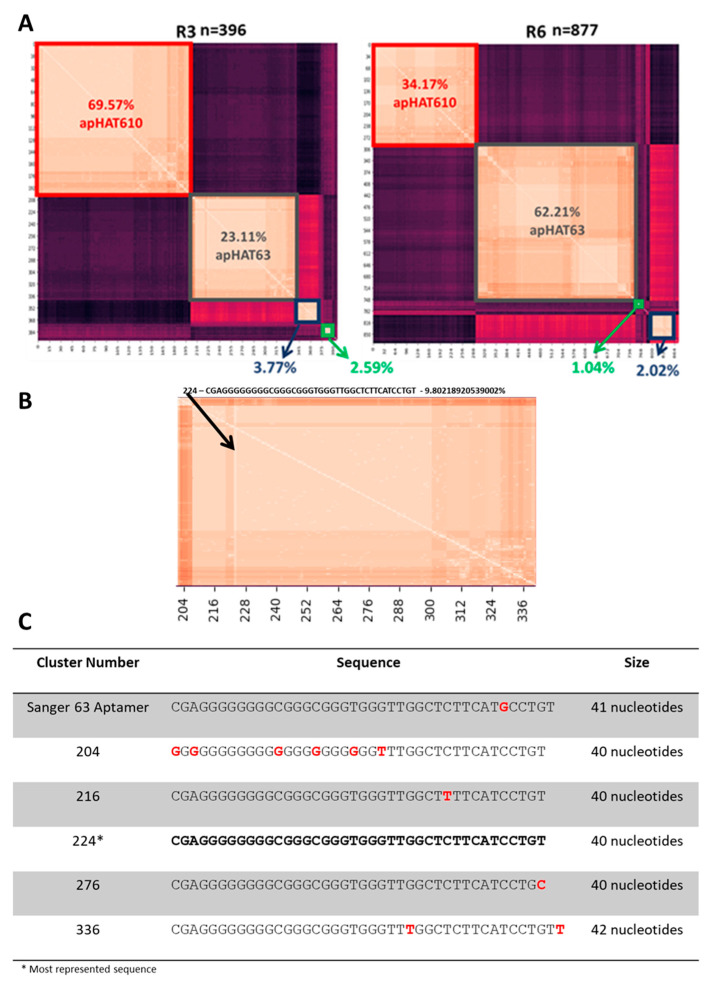
NGS analysis of rounds three and six of aptamer selection against HAT1. (**A**) Heat map of rounds three (**left**) and six (**right**) sequenced with NGS and analyzed with Phyton software according to their sequence similarity. Lighter colors indicate high similarity and darker colors indicate low similarity. The percentage of representation of each cluster was calculated by summing all of the percentages of representation of each sequence. Sequences in the red cluster correspond to apHAT610-like sequences, and sequences in the gray cluster correspond to apHAT63-like sequences that were seen in Sanger sequencing. (**B**) Representative example of the heat map of round three corresponding to the apHAT63 sequence cluster. The lighter color corresponds to the aptamer with the highest similarity to all other aptamers in this cluster. The arrow points to aptamer 224 and indicates the sequence with the lightest color in this cluster, which coincides with the most represented aptamer in the cluster. (**C**) Table showing four randomly chosen sequences representative of the cluster, in addition to the most represented sequence, which is marked with * and bold. The first row shows the sequence of the aptamer obtained by Sanger sequencing. Nucleotides in red indicate existing modifications compared to the most represented aptamer. The sequences shown correspond to the variable region of the aptamers. See also the phylogenetic dendrogram in Appendix A.

**Figure 4 cancers-15-00227-f004:**
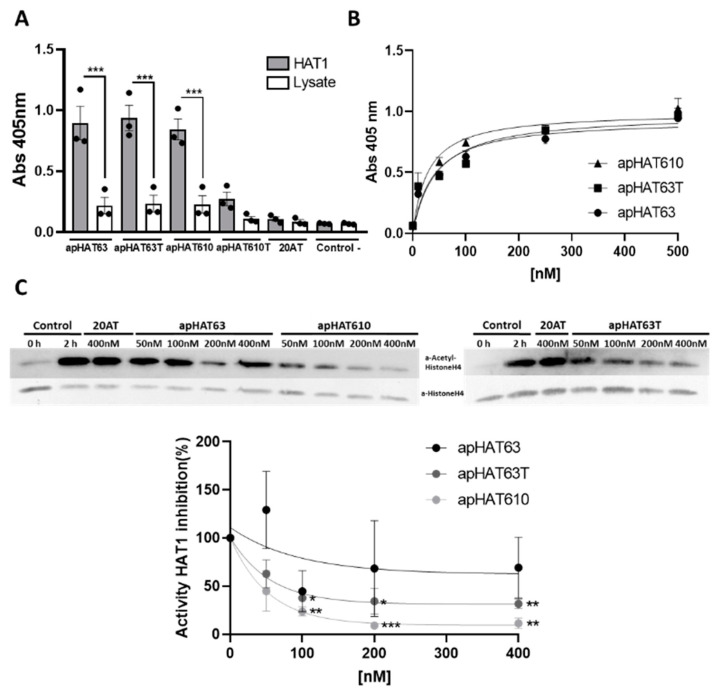
Characterization of the interaction of the selected aptamers with HAT1. (**A**) ELONA of the biotin-labeled synthetic aptamers at 40 nM (100 ng/well) against HAT1 and against a cell lysate previously adsorbed on p96 at 100 ng/well (2.5 pmoL/well). The graph represents the mean ± SEM of three independent experiments (*** *p* < 0.001 with respect to the lysate). (**B**) ELONAs of the biotin-labeled synthetic aptamers at concentrations from 0 to 500 nM versus HAT1 previously adsorbed on p96 at 100 ng/well (2.5 pmoL/well). The graph represents the mean ± SEM of three independent experiments. (**C**) Effect of aptamers on HAT1 activity. Immunodetection with specific antibodies against acetylated histone H4 following separation by 15% SDS-PAGE. Histone H4 levels were used as a loading control. The image corresponds to a representative assay. (Bottom) Quantification of acetylated histone H4 levels normalized according to histone H4 levels in electrophoresis and western blotting. Data are expressed as percentage inhibition with respect to the 2-h control. The graph represents the mean ± SEM of three independent experiments (* *p* < 0.05, ** *p* < 0.01, *** *p* < 0.001 relative to the control).

**Figure 5 cancers-15-00227-f005:**
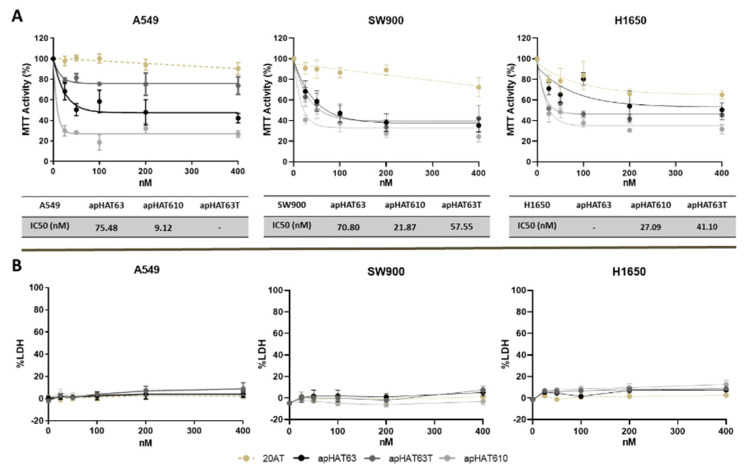
Effect of the aptamers on the MTT and LDH activity of lung cell lines. A549, SW900, and H1650 cells were seeded in p96 at a density of 6 × 10^3^ cells/well for A549 and 10^4^ cells/well for the other two lines. After 24 h, they were transfected with the different aptamers at 25, 50, 100, 200, and 400 nM, and at times of 48 h for A549, 120 h for SW900, and 96 h for H1650 activity assays. (**A**) MTT, with the calculation of the 50% inhibitory concentration (tables located at the bottom of the MTT graphs) and (**B**) LDH were performed. The graphs represent the means ± SEM of three independent experiments.

**Figure 6 cancers-15-00227-f006:**
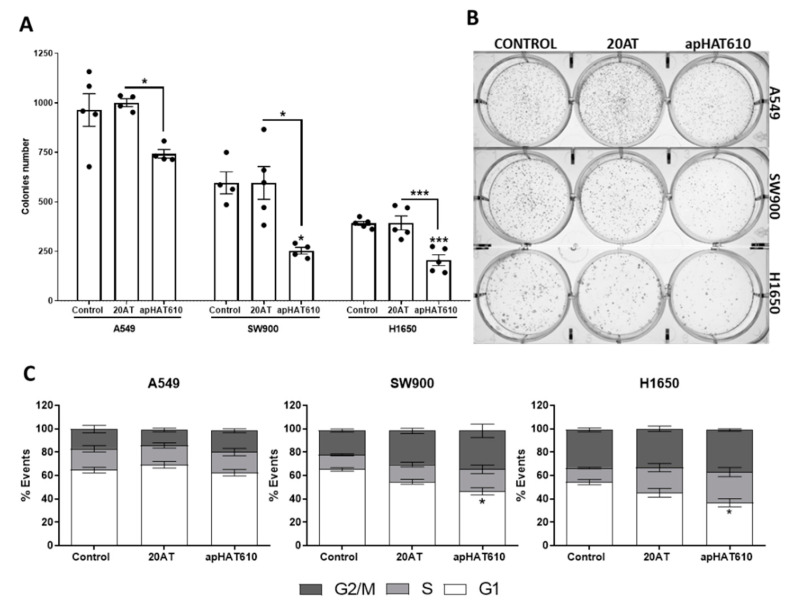
Effect of apHAT610 on the clonogenic capacity and cell cycle in lung tumor cell lines. (**A**) A549, SW900, and H1650 cells were seeded on p6 at density 5 × 10^5^ cells/well and transfected at 24 h with apHAT610 at previously established working concentrations. After 24 h, cells were counted and seeded in p6 at 1000 cells/well for all three lines. The graph represents the mean ± SEM of five independent experiments (* *p* < 0.05 relative to control or relative to 20AT; *** *p* < 0.001 relative to control). (**B**) Representative image of the colonies. (**C**) Cells were seeded at a density of 5 × 10^5^ cells/well in p6, and 24 h later transfected with the apHAT610 aptamer at a concentration of two times the IC50. After 24 h, cells were fixed, stained with propidium iodide, and analyzed by cytometry. Graphs represent the means ± SEM of five independent experiments (* *p* < 0.05 relative to control).

**Figure 7 cancers-15-00227-f007:**
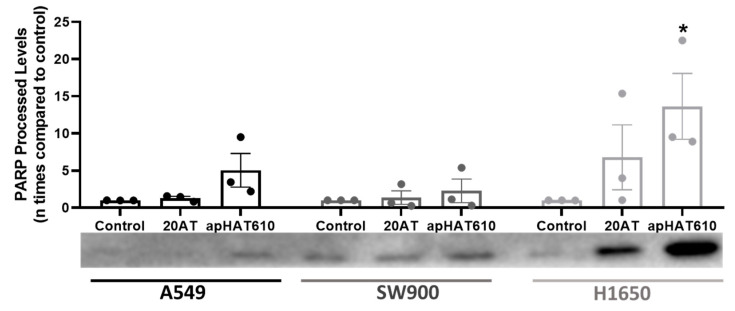
Effect of apHAT610 on apoptosis in lung cell lines. Immunodetection with the specific antibody by 7.5% SDS-PAGE PARP was processed to detect the proteolytic activity of caspase 3 after transfection of A549, SW900, and H1650 lines with apHAT610 and the control aptamer 20AT. The activity of this protease resulted in characteristic proteolysis patterns with specific 89 kDa fragments. The graph represents the mean ± SEM of three independent experiments (* *p* < 0.05 with respect to the control). The image corresponds to a representative assay.

**Figure 8 cancers-15-00227-f008:**
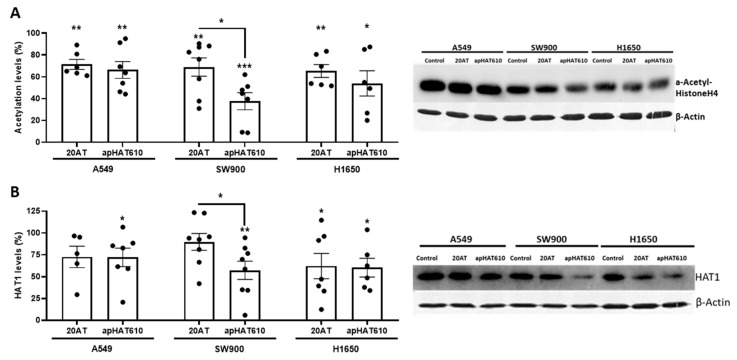
Effect of apHAT610 on histone H4 acetylation levels at 24 h post-transfection. A549, SW900, and H1650 cells were seeded on p6 at density 5 × 10^5^ cells/well and transfected at 24 h with the previously established working concentrations. After the assay time elapsed, the cells were collected and lysed. Subsequently, immunodetection with a specific antibody was performed by 15% SDS-PAGE. (**A**) Histone H4 acetylation levels and (**B**) HAT1 levels at 24 h post-transfection. Data are expressed as the percentage relative to the value in control cells and represent the mean ± SEM of at least eight independent experiments (* *p* < 0.05; ** *p* < 0.01; *** *p* < 0.001). The images on the right correspond to representative experiments from the assays performed with the acetylated histone H4-specific or HAT1-specific antibodies. β-Actin was the loading control.

## Data Availability

All data are shown within the paper.

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
