# Peer review of "Potential Therapeutic Use of Aptamers against HAT1 in Lung Cancer"

_cancers, 2022, doi:10.3390/cancers15010227_

Round 1
Reviewer 1 Report
This work done by Klett-Mingo et al discovered a few new aptamers targeting HAT1 in lung cancer, those aptamers showed potential for the treatment of lung cancer. Below are some suggestions to further improve this work:
“They generate low immunogenicity and toxicity and have high reproducibility since they do not present batch-to-batch variations”: briefly explain why aptamers generate low immunogenicity and toxicity and give proper references.
“the aptamers 63F, 610F, 337 63R, and 610R gave higher absorbance values than the nonspecific control aptamer”: why the complementary strand also show binding affinity toward HAT1? They should have different structural features.
In figure 2, why other 10 enriched sequences show even lower absorbance than the 20AT?
Provide 50 nM, 100 nM, 200 nM conditions for 20AT in figure 4C. why 400 nM apHAT63 show even weaker inhibition effect?
Briefly compare the two different kinds of inhibitors: aptamer inhibitor and chemical inhibitor, and illustrate their advantages and disadvantages.
Characterize the G4 structures formed by apHAT63F and apHAT610R.
Author Response
This work done by Klett-Mingo et al discovered a few new aptamers targeting HAT1 in lung cancer, those aptamers showed potential for the treatment of lung cancer. Below are some suggestions to further improve this work:
“They generate low immunogenicity and toxicity and have high reproducibility since they do not present batch-to-batch variations”: briefly explain why aptamers generate low immunogenicity and toxicity and give proper references.
It is known that aptamers, due to its nucleotide nature, usually show low or no immunogenicity and toxicity. In the text, we indicate references 47 and 58 as an example of articles in which these characteristics of aptamers are indicated. Additionally, we would like to comment that an aptamer against TLR4, developed by our laboratory and the company Aptus Biotech, has shown very low toxicity in phase 1 (First-in-human phase I clinical trial of a TLR4-binding DNA aptamer, ApTOLL: Safety and pharmacokinetics in healthy volunteers. Hernández-Jiménez M, Martín-Vílchez S, Ochoa D, Mejía-Abril G, Román M, Camargo-Mamani P, Luquero-Bueno S, Jilma B, Moro MA, Fernández G, Piñeiro D, Ribó M, González VM, Lizasoain I, Abad-Santos F. Mol Ther Nucleic Acids. 2022 Mar 9;28:124-135) and 2a clinical trials (manuscript in preparation). There are many cases described in the literature with similar results.
“the aptamers 63F, 610F, 337 63R, and 610R gave higher absorbance values than the nonspecific control aptamer”: why the complementary strand also show binding affinity toward HAT1? They should have different structural features.
We agree with the referee. Although the mFold algorithm predicts different structures for the 4 aptamers (Figure S4), the aptamers with complementary sequences have some similarity between them in their secondary structure, which may indicate that this is the structure needed to recognize the target.
On the other hand, aptamers 63F and 610R are very rich in guanines, which would allow them to adopt G-quadruplex structures. Consequently, aptamers with complementary sequences are very rich in cytosines and it has been described that like G-quadruplex DNA structures with intercalated guanine residues, i-motifs consists of antiparallel tracts of oligodeoxynucleotides strands that contain mostly cytosine residues. Consequently, it could be that these structures recognize the same "epitope" as the complementary strand (Gehring, K.; Leroy, J-L; Guéron, M. "A tetrameric DNA structure with protonated cytosine-cytosine base pairs". Nature. 1993 June, 363 (6429): 561–565.).
In figure 2, why other 10 enriched sequences show even lower absorbance than the 20AT?
The aptamers shown in Figure 2 were obtained after cloning and subsequent sequencing of the population enriched in specific aptamers against the target obtained after 6 rounds of SELEX. Within the population one can find aptamers that recognize the target with high affinity and specificity but also other aptamers that can recognize it with lower affinity under the assay conditions. On the other hand, unspecific aptamers may appear in the population that have not been eliminated during the washing processes in the different rounds of selection. In any case, the selected candidates (63F and 610R) have a significantly higher affinity than AT. However, and because the aptamers have been obtained by PCR to conjugate the F or R chain with digoxigenin while the negative control AT used in this assay was a synthetic aptamer, we have modified the figure by removing the AT aptamer to avoid misinterpretation.
Provide 50 nM, 100 nM, 200 nM conditions for 20AT in figure 4C. why 400 nM apHAT63 show even weaker inhibition effect?
In this assay, a 400 nM concentration of 20AT was used since it is the highest concentration used in the other aptamers, because if at that concentration, the 20AT aptamer does not produce any effect in the activity assay, it is to be assumed that it does not do so at lower concentrations either.
On the other hand, it is true that, in the image of inhibition of acetylase activity by the aptamers, apHAT63 at 400 nM shows less inhibition than at 200 nM. However, in the graph analyzing the 3 assays there is no variation in inhibition between 200 nM and 400 nM, although the deviation observed is very high, due to the difficulty of these assays. In any case, the results suggest that apHAT63 does not produce any significant effect on HAT1 activity. A sentence regarding this has been included in lines 484-485.
Briefly compare the two different kinds of inhibitors: aptamer inhibitor and chemical inhibitor, and illustrate their advantages and disadvantages.
We have included a paragraph in the Discussion section (Lines 681-) regarding the comparison between aptamers and chemical HAT1 inhibitors.
Characterize the G4 structures formed by apHAT63F and apHAT610R.
In this work, we only describe the potential of aptamers to form G-quadruplex structures. NMR and circular dichroism assays would be necessary to experimentally characterize whether aptamers actually adopt this type of structures.
Reviewer 2 Report
Introduction does not adequately describe the pro-carcinogenic properties of HAT1 that would make it a compelling target for therapeutic development or delineate the reasons that aptamers are preferred for inhibiting HAT1. The distinction of A- and B-types is not made clear and the relevance of cytoplasmic localization not described. The year of discovery seems irrelevant. The listing of >15 tumor types with elevated HAT1 does not provide a compelling case for the choice of lung cancer for study. Recombinant protein characterization seems incomplete – no MS data for instance. How the aptamer sequences were synthesized and any modifications to confer stability is not described. Cell uptake and sub-cellular localization not described. Spelling errors throughout manuscript decrease reviewer enthusiasm. Most of the results focus on SELEX and aptamer selection that could be published in a nucleic aid focused journal since it is not cancer focused research. More of that data could be included in supplementary results if there were more positive data for the therapy-focused studies. Effects of aptamers on lung cancer cells is not compelling – no clear candidate aptamer with strong general anti-HAT activity for further development. Not clear the findings provide basis for future significant work by this lab or others.
Author Response
Introduction does not adequately describe the pro-carcinogenic properties of HAT1 that would make it a compelling target for therapeutic development or delineate the reasons that aptamers are preferred for inhibiting HAT1.
In addition to the references shown in the introduction (references [9, 29-45] that support the potential of HAT1 as a therapeutic target, we have included a paragraph in which we refer to results obtained in our group that suggest a relevant role of HAT1 in different types of cancer including lung tumors (Lines 112-117).
The distinction of A- and B-types is not made clear and the relevance of cytoplasmic localization not described.
A paragraph has been included in the Introduction section as suggested by the referee (lines 70-88).
The year of discovery seems irrelevant.
The year of discovery has been removed as suggested by the referee.
The listing of >15 tumor types with elevated HAT1 does not provide a compelling case for the choice of lung cancer for study.
Although it is true that the lung is not one of the tumor types in which a higher expression of HAT1 is observed, preliminary results obtained in our laboratory show that there is an increase in the expression of this protein in lung cancer with respect to normal tissue that is greater in advanced stages of the disease and in tumor types with worse prognosis.
On the other hand, in our laboratory we are working on a line of research related to the development of aptamers as potential antitumor drugs in lung cancer, so we decided to use this tumor model. In any case, we are considering validating this aptamer in other tumors in which a higher expression of HAT1 has been observed.
Recombinant protein characterization seems incomplete – no MS data for instance.
The gene coding for HAT1 was cloned into an expression plasmid and the correct sequence was confirmed by Sanger sequencing. The HAT1 protein was then expressed and purified as indicated in the materials and methods section. We have included its characterization, in terms of molecular weight and purity as Supplementary Figure S1. In addition, the results shown in Figure 4 indicate that this protein has acetylase activity, demonstrating that it is in its native configuration.
How the aptamer sequences were synthesized and any modifications to confer stability is not described.
The aptamers were chemically synthesized by IBA Life Sciences as indicated in Materials and methods section. Because of the relatively high stability of the aptamers, probably due to the possibility of their acquiring G-quadruplex structures, we have not included any modification of the aptamers, although this is something that we may raise in the future during preclinical development of these molecules if necessary.
Cell uptake and sub-cellular localization not described.
Relative to cell uptake, the aptamers have been transfected into the cells using Lipofectamine as indicated in Material and methods section. We have performed confocal microscopy studies in two of the cell lines (A549 and SW900) representative of the two tumor types (adenocarcinoma and squamous carcinoma). A sentence has been included in lines 286-287. The results show that aptamers are located in the cytoplasm and the nucleus in both cell lines. The figure has been included al supplementary figure S2.
Spelling errors throughout manuscript decrease reviewer enthusiasm.
The manuscript has been sent to an editing service for English revision.
Most of the results focus on SELEX and aptamer selection that could be published in a nucleic acid focused journal since it is not cancer focused research. More of that data could be included in supplementary results if there were more positive data for the therapy-focused studies.
We have considered that the identification and characterization of aptamers against HAT1 as potential antitumor drugs in lung cancer fits perfectly with the objective of this special issue on aptamers and cancer.
Effects of aptamers on lung cancer cells is not compelling – no clear candidate aptamer with strong general anti-HAT activity for further development. Not clear the findings provide basis for future significant work by this lab or others.
In accordance with the referees' suggestions, we have repeated some assays in order to support the data more firmly. In view of the results obtained, we consider the apHAT610 aptamer, which shows better IC50 values than the HAT1 inhibitors published so far, to be sufficiently robust to continue with the further development of the apHAT610 aptamer as a potential antitumor agent in lung cancer and other tumors.
Reviewer 3 Report
In this manuscript, authors describe the selection of aptamers against the Histone acetyltransferase 1 (HAT1), as potential drugs for the treatment of lung cancer.
In my opinion, the manuscript in this form is not acceptable for publication. Some major concerns need to be importantly addressed to make the paper suitable for publication.
Major concerns:
- Authors should show, also in supplementary figures if they prefer, the phylogenetic dendrograms related to sequences obtained by Sanger and Next Generation Sequencing.
- In Figure 4C, the reported results appear to come from 3 different gels. The gels for apHAT63T and apHAT610 are devoid of the controls present in the apHAT63 gel. The presence of these controls on each gel is necessary for the correct analysis and interpretation of the results. Again, do the authors have an explanation for the not perfectly dose dependent pattern observed for the apHAT63?
- Results showed in figure 6A are not very convincing. Indeed, to be confident about aptamer effect on cell clonogenic capacity, results should be statistically significant with respect to both control and 20AT. Thus, the results shown in this way would indicate a compelling effect only on H1650 cells. In any case, I suggest the authors to repeat the experiments in more replicates, carefully reviewing the protocol used.
- Also results on cell cycle (Figure 6C) and histone H4 acetylation (Figure 8) seem to be unclear. Indeed, also in this case the behavior of the 20AT control aptamer in most cases seems to question a specific effect of the aptamer. Again, I would recommend repeating the experiment in more replicates, re-evaluating the protocol used, and considering the use of a different control aptamer in addition to the AT20, in this and also in the other experiments of the paper.
Minor concerns:
- Please, indicate the fold increase in Figure 1 A
- Please, indicate samples’ name in Figure 6B
Author Response
In this manuscript, authors describe the selection of aptamers against the Histone acetyltransferase 1 (HAT1), as potential drugs for the treatment of lung cancer.
In my opinion, the manuscript in this form is not acceptable for publication. Some major concerns need to be importantly addressed to make the paper suitable for publication.
Major concerns:
- Authors should show, also in supplementary figures if they prefer, the phylogenetic dendrograms related to sequences obtained by Sanger and Next Generation Sequencing.
We have made a phylogenetic dendrogram analyzing the 24 most represented sequences in NGS and the sequences obtained by Sanger sequencing. The results have been included as Figure S3 in Supplementary Material.
- In Figure 4C, the reported results appear to come from 3 different gels. The gels for apHAT63T and apHAT610 are devoid of the controls present in the apHAT63 gel. The presence of these controls on each gel is necessary for the correct analysis and interpretation of the results. Again, do the authors have an explanation for the not perfectly dose dependent pattern observed for the apHAT63?
Figure 4C has been modified to include the complete gels as suggested by the referee.
It is true that, in the image of inhibition of acetylase activity by the aptamers, apHAT63 at 400 nM shows less inhibition than at 200 nM. However, in the graph analyzing the 3 assays there is no variation in inhibition between 200 nM and 400 nM, although the deviation observed is very high, due to the difficulty of these assays. In any case, the results suggest that apHAT63 does not produce any significant effect on HAT1 activity. A sentence regarding this has been included in lines 472-474.
- Results showed in figure 6A are not very convincing. Indeed, to be confident about aptamer effect on cell clonogenic capacity, results should be statistically significant with respect to both control and 20AT. Thus, the results shown in this way would indicate a compelling effect only on H1650 cells. In any case, I suggest the authors to repeat the experiments in more replicates, carefully reviewing the protocol used.
- Also results on cell cycle (Figure 6C) and histone H4 acetylation (Figure 8) seem to be unclear. Indeed, also in this case the behavior of the 20AT control aptamer in most cases seems to question a specific effect of the aptamer. Again, I would recommend repeating the experiment in more replicates, re-evaluating the protocol used, and considering the use of a different control aptamer in addition to the AT20, in this and also in the other experiments of the paper.
As suggested by the referee, we have performed new clonogenic activity and cell cycle experiments and histone H4 acetylation assays in vivo. The results are shown in the new Figures 6 and 8. In addition, outlier`s analysis has been performed in the statistical studies.
With respect to the 20AT control, we do not observe any effect on the clonogenic activity and cell cycle after including the data obtained in the new experiments. In the case of H4 acetylation assays, a substantial reduction in HAT1 acetylase activity is observed when we treat all 3 cell lines with apHAT610, being particularly significant in SW900 cell line. We consider that the inhibition exerted by 20AT control (around 30% in the three cell lines) would be a nonspecific effect since 20AT does not inhibit HAT1 activity in the in vitro inhibition assays.
Minor concerns:
- Please, indicate the fold increase in Figure 1A
Fold increase has been included in lines 333-334.
- Please, indicate samples’ name in Figure 6B
Figure 6B has been modified to include the samples' name as suggested by the referee.
Round 2
Reviewer 1 Report
suggest acceptence
